# Care for Patients with Type-2 Chronic Rhinosinusitis

**DOI:** 10.3390/jpm13040618

**Published:** 2023-03-31

**Authors:** Gianmarco Giunta, Francesca Pirola, Francesco Giombi, Giovanna Muci, Gian Marco Pace, Enrico Heffler, Giovanni Paoletti, Francesca Puggioni, Michele Cerasuolo, Fabio Ferreli, Fabrizio Salamanca, Giuseppe Mercante, Giuseppe Spriano, Giorgio Walter Canonica, Luca Malvezzi

**Affiliations:** 1Department of Biomedical Sciences, Humanitas University, Via Rita Levi Montalcini 4, Pieve Emanuele, 20090 Milan, Italy; 2Otorhinolaryngology Head & Neck Surgery Unit, IRCCS Humanitas Research Hospital, Via Manzoni 56, Rozzano, 20089 Milan, Italy; 3Personalized Medicine, Asthma and Allergy, IRCCS Humanitas Research Hospital, Via Manzoni 56, Rozzano, 20089 Milan, Italy; 4Otorhinolaryngology Head & Neck Surgery Unit, Casa di Cura Humanitas San Pio X, Via Francesco Nava 31, 20159 Milan, Italy

**Keywords:** chronic rhinosinusitis, nasal polyps, personalized medicine, monoclonal antibodies, type-2 inflammation, sinus surgery

## Abstract

In the last 20 years, growing interest in chronic rhinosinusitis (CRS) has become evident in medical literature; nevertheless, it is still difficult to identify the real prevalence of the disease. Epidemiological studies are few and focused on heterogeneous populations and diagnostic methods. Recent research has contributed to identifying CRS as a disease characterized by heterogeneous clinical scenarios, high impact on quality of life, and elevated social costs. Patient stratification with phenotypes and identification of the pathobiological mechanism at the origin of the disease (endotype) and its comorbidities are pivotal in the diagnostic process, and they should be addressed in order to properly tailor treatment. A multidisciplinary approach, shared diagnostic and therapeutic data, and follow-up processes are therefore necessary. Oncological multidisciplinary boards offer models to imitate in accordance with the principles of precision medicine: tracing a diagnostic pathway with the purpose of identifying the patient’s immunological profile, monitoring therapeutical processes, abstaining from having only a single specialist involved in treatment, and placing the patient at the center of the therapeutic plan. Awareness and participation from the patient’s perspective are fundamental steps to optimize the clinical course, improve quality of life, and reduce the socioeconomic burden.

## 1. Introduction

Chronic rhinosinusitis (CRS) is defined as a long-lasting (>12 weeks) inflammation of the nasal cavity and paranasal sinuses, characterized by symptoms of nasal blockage/congestion or nasal discharge, possibly associated with facial pain/pressure and a dysregulated sense of smell [1]. Yet, careful evaluation and subsequent distinction between the different types of pain in that precise area of the body must be made. In this regard, an apparently common symptom such as sinus headache and/or facial pain often harbors a serious diagnostic challenge. Traditionally, rhinogenic pain is unilateral, severe, located on the same side as and related to rhinologic symptoms, and almost always accompanied by endoscopic and CT abnormalities. However, neurogenic pain may also be perceived in the maxillofacial region and, likewise, may be associated with congestion and nasal discharge (e.g., a cluster headache). Furthermore, current evidence suggests that sinus and/or facial complaints should be attributed in most cases to primary headaches (e.g., migraines and tension-type headaches) [2,3,4]. For these reasons, neurological pain should be ruled out before proceeding to a multidisciplinary consultation, as we propose in this paper, that is rhino-allergist in conduction. This is of crucial importance, both to avoid wasted time for the patient, who may suffer from frustration caused by incorrect care and years of reduced quality of life, and to avoid loss of resources for the healthcare facility.

CRS is a generic term that may be useful to establish diagnosis but widely inadequate in defining the complexity of differing clinical patterns of the disease.

During the last 40 years, clinicians and researchers have underlined the importance of considering the subjective dimension of the disease to achieve a more holistic overview of the patient and of the effects of the whole healthcare process. This point of view was driven by the clinical necessity to shift from “disease-centered medicine” to a more global perspective of “patient-centered medicine” [5]. Therefore, any information directly provided by the patient about a health condition and its management (Patient Reported Outcomes—PROs) represents a fundamental component of any treatment paradigm aimed to provide a personalized approach [6]. In 1948, the World Health Organization defined “health” as a state of complete physical, mental, emotional, and social well-being and not merely the absence of disease or infirmity [7]. Accordingly, physicians need to be aware that patients with type-2 inflammatory diseases of the respiratory tract should be addressed from a patient-centered perspective.

There is a growing interest in literature on the high socioeconomic impact (direct and indirect costs on society) and reduced quality of life (QoL) of CRS, as also abundantly underlined in the European Position Paper on Rhinosinusitis and Nasal Polyps (EPOS) [1]. CRS with nasal polyps (CRSwNP) is relatively common, particularly in asthmatics, affecting about 2–4% of the general population [8,9] but with increasing prevalence among unselected asthmatics (7–15%) [10] and affecting up to 50% of patients with severe asthma, particularly those with severe late-onset eosinophilic asthma [11]. The economic and social burden of rhinosinusitis, both acute and chronic, is enormous [12]. Costs of medication, hospitalization, physician’s examinations, and surgery account only for direct healthcare expenses, while there is a concurrent and similar substantial indirect cost from absenteeism, disability, and therefore loss in productivity and work performance. According to SF-36 and other health-related QoL measures, absenteeism and decreased quality of life are particularly linked to some forms of rhinitis, such as recurrent acute rhinosinusitis (RARS) and CRS both with and without polyps (CRSsNP), and also with high prevalence (15–25%) of related depression and anxiety [13]. In the USA, rhinosinusitis is one of the “top 10” costliest health conditions for employers. The current direct costs for the management of CRS are between $10 and $13 billion per year, with the highest direct costs found for patients who had recurrent polyposis after surgery. Indirect costs from absenteeism and presenteeism (decreased productivity at work) significantly add further economic burden. Overall, just in the USA, the total indirect costs related to CRS were estimated to be $20 billion per year [1].

Moreover, the correlation between CRS and asthma eventually amplifies the burden of both conditions synergistically [14]. Indeed, the concept of rhino-bronchial syndrome [15], which has been introduced to highlight the link between upper and lower airway pathophysiology, is widely accepted.

President Obama, in his 2015 Precision Medicine Initiative, defined precision medicine as “a bold new research effort to revolutionize how we improve health and treat disease.”. Precision medicine goes beyond the one-size-fits-all approach, considering individual differences in people’s genes, environments, and lifestyles. This concept was widely introduced with the 4P medicine paradigm, where 4P stands for prediction, prevention, personalization, and participation. The first three Ps were introduced at the beginning of the century, and the fourth one was added on in 2014 by the molecular biologist and oncologist Leroy Hood [16]. This extension has been labeled as “a driving force for revolutionizing healthcare,” since the individual’s participation is the key to putting into practice the other three aspects [17,18].

The same 4P paradigm has been applied to CRS [19,20]. Since CRS is a chronic disorder, the primary fact is that medicine cannot cure CRS patients. Still, it has the ability to improve its course and lower the impact on QoL and on social costs and is also a means of predicting—hence avoiding—possible undesirable progression and maintaining wellness (prevention). Participation is fundamental: keeping the patient at the core of the treatment plan and encouraging counseling to maintain adherence and compliance. Overall, as the response to treatment is influenced by several factors, patient stratification is fundamental to set the correct diagnostic and therapeutic course for each patient. Identifying markers that may be predictive of response means actualizing the concept of target therapy (precision) and predicting the response to it (prediction). Based on the model adopted for oncological patients, physicians should establish a multidisciplinary team to plan the correct personalized treatment for CRS patients (Figure 1). Therefore, as with all the chronic conditions that impact quality of life, an individualized care plan is needed for diagnosis, treatment, and long-term observation.

## 2. Discussion

Etiopathological mechanisms

Despite being clearly distinguishable, CRS phenotypes do not provide full knowledge of the underlying cellular and molecular pathophysiologic mechanisms of the disease, which are increasingly relevant because of the different associations with comorbidities and their responsiveness to treatments such as corticosteroids, surgery, and/or biological agents [21]. Indeed, CRS etiology is very complex to define. At the basis of CRSwNP, in Western countries, rests the so-called type-2 inflammation and related cytokines. Type-2 immune responses are defined by interleukin-4 (IL-4), IL-5, IL-9, and IL-13, which can be host protective yet, when dysregulated, have pathogenic activity [22]. Type-2 immunity induces a complex response involving granulocytes (eosinophils, basophils), mastocytes, type-2 innate lymphoid cells (ILC2), IL-4- and/or IL-13-conditioned macrophages and T helper 2 (Th2) cells. These cells are crucial to the pathogenesis of CRS and related disorders and are therefore driving mechanisms that control the intensity, maintenance, and resolution of type-2 immunity. They are also reasonably important regulators of disease progression and must be fully understood for therapeutic purposes.

In the context of damaged airway epithelium as in CRSwNP, triggers that initiate and perpetuate type-2 inflammation may be different: viruses, bacteria, allergens, cigarette smoke, pollutants, etc. Yet, there is increasing evidence that viruses may enhance type-2 immunological response by stimulating the epithelial expression of the cytokines thymic stromal lymphopoietin (TSLP), IL-25, and IL-33. In turn, these cytokines promote an intense cellular response by activating the mast cells ILC2 and Th2 that release IL-5, IL-13, and IL-4. The result of this immune cascade is the activation of eosinophils (by means of IL-5 and IL-13) and B-cells (through IL-4). IL-4 and IL-13 and their common receptor complex (IL-4Rα) are significantly elevated in CRSwNP. Despite sharing the same signaling pathway, they play distinct roles: IL-4 is mainly involved in the survival and proliferation of Th2 cells and isotype class switching of B cells to produce IgE, while IL-13 induces airway hyperreactivity and mucus hyperproduction, as well as smooth muscle proliferation and fibrosis (i.e., tissue remodeling). IL-4 and IL-13 induce the abovementioned conditioned state of macrophages, a typical feature of CRSwNP [23,24].

Another molecule that is implicated in the pathogenesis of both CRSsNP and CRSwNP is transforming growth factor-β (TGF-β). TGF-β signals are fundamental in safe-guarding the functions of specific regulatory T cells (Treg) [25]. As Treg cells and TGF-β pathway are critical regulators of T-cell tolerance, together they play important roles in the development of immune disorders, such as asthma and allergies [26]. In addition, TGF-β is a key factor in the remodeling process found in sinonasal mucosa with CRS. Specifically, TGF-β pathways were found to be upregulated in CRSsNP and downregulated in CRSwNP [27]. In the former scenario, TGF-β upregulation leads to the induction of proliferation of fibroblasts, increased collagen deposition, and extracellular matrix (ECM) production, as well as reduced metalloproteinases (MMP) activity, hence resulting in fibrosis and basement membrane thickening [28]. In CRSwNP, TGF-β downregulation contributes to greater MMP activity, degradation of ECM, and deposition of albumin, which results in intense edematous stroma, subepithelial and perivascular inflammatory cell infiltration, formation of pseudocysts, and polypoid degeneration.

Genetical predisposition seems to play a role in the pathogenesis of the disease. In a recent study of six hundred patients affected by CRSwNP, the overall prevalence of a familial link has been estimated to be 29.5%, with a significantly higher prevalence in first-degree compared to second-degree relatives [29]. Furthermore, a large database study suggested a significant familial risk associated with pediatric CRS [30], although studies on monozygotic twins have shown that siblings do not develop polyps simultaneously, indicating that environmental factors are as likely as genetic ones to influence the occurrence of nasal polyps [1,31]. Associations of CRS and polymorphisms in more than 30 genes have been published, with single nucleotide polymorphisms in 3 of them (IL-1α, TNF-α, Acyloxyacyl hydrolase) [32]. Supportive evidence for a genetic basis of CRS also includes the fact that several syndromes associated with known genetic defects have CRS as a clinical component. This includes primary ciliary dyskinesia [33] as well as monogenic diseases such as cystic fibrosis, which is caused by a deficiency in epithelial chloride transport due to mutations in the cystic fibrosis transmembrane conductance regulator (CFTR) gene [34].

Bacterial colonization, which may conduce to impaired mucociliary function, also plays a role in the onset or maintenance of the inflammatory process in CRS. In a healthy state, commonly identified bacterial strains in the upper airways include *Staphylococcus*, *Corynebacterium*, *Peptoniphilus*, and *Propionibacterium*, among others [35]. Interestingly, the total bacterial load in healthy and diseased sinuses appears to be surprisingly similar across adults. Accordingly, current evidence suggests that the lack of host response to normal sinus flora may be critical for the development of CRS, leading to the proliferation of opportunistic pathogens [36]. The phenomenon of antibiotic resistance, which is often observed in refractory CRS, is still not fully understood: recent research is strongly focused on the role of the bacterial biofilm, a three-dimensional structure composed of a mixture of biopolymers, primarily polysaccharides, which provides a mechanism for increased survival, protecting bacteria against antibodies, phagocytosis, antibiotic penetration, and complement binding [37,38]. In a biofilm community, bacteria have been estimated to be up to 1000 times more resistant to antibiotic therapy [39]. Even though it is not yet applicable to clinical practice, by investigating the mechanism of biofilm resistance, future research will be aimed at overcoming drug resistance, thus providing new therapeutical targets in refractory CRS.

Further mechanisms underlying CRS are complex and likely still unknown. As phenotyping is sometimes found insufficient for treatment efficacy, endotyping becomes necessary. In the scenario of precision medicine, identifying the specific pathophysiologic process of a patient’s endotype may permit more effective treatments, better patient outcomes, and lower expenditures [40]. In addition, defining the endotype is required to access the use of new targeted biotherapeutic molecules, such as anti-IgE and anti-cytokine monoclonal antibodies [41].

Surgical management: contemporary and future perspectives

Since 1986, functional endoscopic sinus surgery (FESS) has become the worldwide gold standard in the management of sinonasal inflammatory disease that is unresponsive to medical therapy. This procedure, as it was originally presented, currently leaves some interpretative doubts. According to the original principles as described by Stammberger, the aim of FESS is the rehabilitation of proper sinus ventilation by creating a sinus cavity that incorporates the natural ostium without altering its patterns of drainage, in order to promote the proper mucociliary clearance and, eventually, to facilitate the instillation of topical therapies [42]. The idea behind functional surgery encompasses minimizing mucosal stripping and preserving anatomical landmarks (such as the middle turbinate). Despite the fact that future perspectives are based on developing new drugs aimed to target the abovementioned molecular pathways to prevent recurrent sinus surgery or even avoid it, nowadays steroid therapy and surgery still play a relevant role in the management of CRS. Anti-inflammatory therapies are at the forefront of the treatment of eosinophilic CRSwNP (eCRSwNP) and, above all, corticosteroids, both intranasal and systemic. They contrast with type-2 inflammation, thus controlling both local and associated systemic effects of the disease [43]. However, chronic and/or recurrent use of systemic corticosteroids (particularly frequent in patients with CRSwNP and concomitant severe asthma [44] is associated with a relevant increased risk of developing adverse conditions (e.g., type-2 diabetes, hypertension, glaucoma, osteoporosis) [45], significantly contributing to the socioeconomic impact of the disease [46].

Along with steroids, surgery is indicated to relieve nasal symptoms in patients with CRS. Functional surgery (i.e., FESS) specifically means to clear ethmoidal clefts, reestablishing ventilation and drainage of diseased large sinuses via their physiological routes: the frontal recess for frontal sinus ventilation and the lateral wall of the middle meatus for maxillary sinus ventilation. Minor deviations from the FESS paradigm take place, for example, when the maxillary sinus ostium is enlarged anteriorly and/or posteriorly (to the nasal fontanelle areas), still resulting in a window in its physiological place. Over time, the concept of FESS has developed further. According to the EPOS2020 steering group, “full FESS” indicates sinus opening that includes anterior and posterior ethmoidectomy, large middle meatal antrostomies, sphenoidotomy, and frontal opening (e.g., a Draf IIa procedure), still without damaging important landmarks like the middle turbinate and the mucosa in general [1]. This is particularly applicable to compartmental sinusitis and CRSsNP, as well as to non-type-2 CRSwNP. The functionality criterion should also be respected in severe CRSwNP and in conditions characterized by type-2 inflammation. However, in a selected group of patients characterized by a significant mucosal inflammatory load, limited functional surgery may not be effective in the long run [47]. Nasal polyposis is histologically dominated by the presence of polymorphonuclear leukocytes: in cases where the pattern of nasal mucosa infiltration is highly predominant in eosinophils, the disease presents a more severe course; thus, an aggressive surgical approach may be justified. [48,49,50,51]. Moreover, in order to prevent polyps’ regrowth and to preserve adequate clinical outcomes, surgery is often combined with extensive postoperative use of corticosteroids [52]. As the disease becomes more severe, wider surgical resections end up being necessary: a large ethmoidectomy box with wide lateral fenestration to the maxillary sinus, extended upward to the frontal sinus and backward to the sphenoid. In many cases, the steadiness of the middle turbinate is compromised by both the destructive action of the disease and the extension of surgery. Being systematic in the endoscopic approach (ESS), implementation of designing a targeted surgical treatment responds to the following needs: to create a surgical field as wide as possible to help control recurrence with topical medications, to minimize post-surgical restenosis, and to facilitate eventual re-intervention by simple polyp debridement.

In 2018, a newly proposed approach (named the “reboot approach”) [53,54] was introduced in the surgical scenario of severe recalcitrant CRSwNP cases, especially for patients who underwent multiple interventions. Reboot surgery aims to restore a non-inflammatory state of the epithelium by entirely removing the dysfunctional eosinophilic-infiltrated mucosa up to the periosteum of nasal and paranasal cavities, partially sparing the mucosa of the inferior conchas. The procedure is accompanied by Draf III or at least Draf IIb frontal drainage. The rationale is that removal of the type-2 inflammatory environment might allow unaffected mucosa to grow and re-epithelize sinus walls, markedly decreasing the risk of relapse. The surgical procedure consists of the clearance of all polyps and diseased mucosa down to the periosteum of all paranasal sinuses. A large antrostomy and the use of angled instruments guarantee the correct exposure of all maxillary walls, and diseased mucosa is removed from the periosteum of the anterior wall and alveolar recess. Similarly, removal of sinonasal mucosa is extended to the sphenoid ostium, paying particular attention to lateral and superior borders to avoid damage to the carotid sheet or the optic nerve. Reboot surgery, as described by Alsharif et al., is classified as full or partial, depending on whether a Draf III frontal senotomy is performed [53]. The removal of diseased mucosa, in a particular subset of patients affected by recalcitrant drug-resisting type-2 CRSwNP, has recently been shown to allow restoration of infiltrate-free epithelium, improving recurrence-free survival, quality-of-life outcomes, and the need for oral corticosteroid uptake [55].

Despite great advancements in the field of targeted molecular therapy, nowadays the indications for monoclonal antibodies are still often strict, due to the high social costs and the relatively still limited diffusion worldwide [56]. In this context, surgery should not be overwhelmed by medical targeted therapy but rather complete it from the perspective of a personalized approach to the patient.

Accordingly, an upcoming fascinating perspective in the future could be represented by the combination of surgery and biological therapy for eCRSwNP. In this manner, endoscopic surgery could be minimized to the true principles of FESS, supported by the effects of post-operative administration of targeted drugs.

Care for pediatric patients with chronic rhinosinusitis

The issue of pediatric CRS was addressed in EPOS2020, since it exhibits peculiar characteristics [1]. In the United States, chronic rhinosinusitis accounts for 5.6 million visits per annum (range, 3.7–7.5 million) among patients 0 to 20 years of age, being diagnosed in 2.1% of the pediatric population who present in the outpatient setting [57]. Although many of the inflammatory markers are similar to the abovementioned markers observed in adults, evidence concerning the etiology of pediatric CRS is very heterogeneous, thus not allowing the same process of endotyping. The higher prevalence of CRS in patients affected by hereditary syndromes such as cystic fibrosis or primary ciliary dyskinesia highlights the role of mucociliary clearance impairment in the development of the disease [33,34]. Furthermore, a critical role in the pathogenesis of pediatric CRS could be played by the lymphoreticular adenoid tissue, which may act as both a reservoir for pathogenic bacteria and a source of obstruction, thus causing impaired mucociliary clearance of the sinus cavities [58,59]. Several studies have been performed to evaluate the relationship between the adenoid pad and paranasal sinus in pediatric patients with CRS. Arnaoutakis et al. used Andersen’s saccharine test in 10 patients with adenoid hypertrophy to measure the nasal mucociliary clearance time (MCT) and mucociliary velocity (MCV) before and after adenoidectomy. They found that both MCT and MCV improved postoperatively among the group, although the results were still not statistically significant due to the limited sample size [60].

As in adults, the diagnostic process includes an accurate anamnestic collection, which, however, should not ignore the parents’ interview. In this regard, from the perspective of a holistic approach and in line with the definition of health given by the WHO, one should also consider that the greatest impairments to well-being in children with CRS were estimated to be the impact of the child’s health status on parents’ emotions, pain and discomfort, and general perception of health [61]. In order to confirm the diagnosis, nasal endoscopy and eventual computed tomography (CT) are justified and recommended by current guidelines, keeping in mind the added risk of exposure to radiation [1,62].

Topical therapy is considered the first-line strategy for pediatric CRS [1]. Once-daily intranasal irrigation for 6 weeks using saline has demonstrated a safer profile and no significant differences in clinical outcomes compared to a group also using an antibiotic (gentamicin) and is therefore recommended for pediatric CRS [63]. Evidence regarding the efficacy of intranasal steroids in the treatment of CRS in children is weak. Only one randomized controlled clinical trial has been performed evaluating intranasal steroids in 127 children with CRS with nasal polyposis. The intent of this study was to provide evidence for the safety of intranasal mometasone, but the authors also reported that improvement in congestion did not reach statistical significance [64]. Therefore, even in light of the favorable efficacy in adults with CRS and children with rhinitis, EPOS2020 also extended the recommendation for use of intranasal steroids in pediatric CRS patients [65,66]. Conversely, there is no current indication for the administration of either oral or intravenous antibiotics, due to lack of supportive evidence [1]. Surgery may be considered for refractory or complicated patients. Adenoidectomy, considering its relative efficacy and safety, has been addressed as the first-line surgical treatment for pediatric CRS in EPOS2020 [1] in order to remove the bacterial reservoir observed at this level and to reestablish a physiological mucociliary clearance pattern. Typically, maxillary sinus irrigation is also performed concurrently, and this is specifically recommended in patients with a high Lund-Mackay score following a CT scan [1]. More extensive sinus surgery should be considered for children who fail adenoidectomy and antral washout, or for children with more severe diseases (e.g., primary ciliary dyskinesia, cystic fibrosis). In this regard, one must remember that pediatric anatomy conceals many peculiarities. Paranasal sinuses in children, for instance, continue to develop until early adulthood and reach maturity at different rates, with the frontal sinus being the last to pneumatize. A final paranasal sinus conformation is reached by 12 years or older [67,68]. However, evidence support FESS as a second-line treatment for refractory pediatric CRS and in older children [1]. A prospective observational study of 39 children affected by CRSwNP (mean age 10.9 years) undergoing ESS demonstrated an improvement in quality of life after surgery [69]. Likewise, another non-randomized controlled prospective study showed that SNOT-22, Lund-Mackay, and Lund-Kennedy scores improved after ESS in 132 children with pediatric CRS compared to 15 healthy controls [70].

In conclusion, in the era of personalized medicine, children should no longer be considered as “smaller adults” [71], because of the many differences, not only from a biological and anthropometric point of view but also from an emotional perspective. In order to offer a tailored approach to the child, the inclusion of the pediatrician in the multidisciplinary consultation should be advised.

Are biologics grabbing the spotlight?

As vastly proven, initial treatment of CRS includes topical and systemic corticosteroids, long-term antibiotics, and surgical intervention. As mentioned, however, some patients may suffer from a recalcitrant form of disease despite best practices. In recent years, the advancement in pharmaceutical therapies has also reached application in CRSwNP, with molecules (i.e., monoclonal antibodies) that specifically target the major players in the inflammation cascade of this condition. With regard to the use of biologics, the European Forum for Research and Education in Allergy and Airway Diseases (EUFOREA) suggested five criteria that should be satisfied to prescribe them for CRSwNP [72]. Besides having undergone sinus surgery, at least three criteria must be met among the following: the evidence of type-2 inflammation, the need for systemic corticosteroids in the past two years, a significant impairment in quality of life because of the disease, reduced sense of smell, and comorbid asthma. The idea that biologics may become an alternative to surgery is still a matter of debate that will eventually be ascertained during the following years, after their definite approval for CRSwNP and the post-marketing surveillance phase.

The critical players that have been targeted so far are IgE, IL-5, IL-4, and IL-13, as well as some of their receptors.

The continuous local activation by the IgE-receptor pathway of mast cells, basophils, and dendritic cells can be reduced by selective binding to free circulating IgE, thanks to an anti-IgE antibody called omalizumab (Xolair^®^). Two pivotal early studies [73,74] in a small number of patients (<30) gave contradictory results, but, very recently, the initial results of two phase-III clinical trials (POLYP 1 and POLYP 2) showed that omalizumab significantly improved endoscopic, clinical, and patient-reported outcomes in patients with corticosteroid-refractory CRSwNP [75]. These were two randomized, multicenter, placebo-controlled trials whose results showed a significant improvement in NPS, SNOT-22, and University of Pennsylvania Smell Identification Test scores for omalizumab versus a placebo.

In addition, two drugs that act by blocking circulating IL-5 have been developed: mepolizumab (Nucala^®^) and reslizumab (Cinqaero^®^), though the latter has still not been approved in Italy. These monoclonal antibodies, which interrupt eosinophilic inflammation, have undergone testing through randomized controlled studies: the only large study that was conducted with 105 severe CRSwNP patients shows that mepolizumab reduced the need for sinonasal surgery compared to a placebo [76]. Significant improvements are manifest in symptoms (rhinorrhea, nasal blockage, and hyposmia), quality of life (by means of the Sino-nasal Outcome Test, SNOT-22, questionnaire), and increase in peak nasal inspiratory flow (PNIF). Similarly, blood eosinophils decrease. Phase-III trials for mepolizumab in CRSwNP are in progress. Mepolizumab administration has already been approved for severe eosinophilic asthma as well as for eosinophilic granulomatosis with polyangiitis or hypereosinophilic syndrome. The strongest evidence supporting its use in type-2 CRSwNP is SYNAPSE, a randomized, double-blind, placebo-controlled, parallel-group phase 3 trial conduced in 93 centers [77]. In SYNAPSE, mepolizumab administration significantly improved NPS and nasal obstruction compared with a placebo. Apart from the proper interleukin, a drug targeting the interleukin receptor exists: benralizumab (Fasenra^®^). Randomized controlled trials on benralizumab, directed towards the alpha subunit of the IL-5 receptor (IL-5Rα), are ongoing. OSTRO was a randomized, double-blind, placebo-controlled, multicentric study that proved that benralizumab significantly improved NPS as well as nasal blockage and sense of smell scores compared to a placebo [78].

Interleukin-4 (IL-4) and interleukin-13 (IL-13) have overlapping biological effects because of their partly shared receptor complex. IL-4 may interact with either a type-I receptor (made of IL-4 receptor alfa, IL-4Rα, and the common γ-chain of the IL-2R) or a type-II receptor (made of IL-4Rα and an IL-13 binding chain, IL-13Rα1). This type-II receptor complex is also a functional receptor for IL-13, which is the reason for IL-4/IL-13 common pathways and functional properties [79]. Dupilumab, an anti-IL-4 receptor alpha (IL-4Rα) monoclonal antibody (mAb), targets IL-4 and IL-13 signaling simultaneously, and its administration has been demonstrated to significantly improve a patient’s nasal polyp score (NPS), nasal congestion or obstruction, and sinus Lund-Mackay (LM) CT scores in two phase-III big trials [80]. SINUS-24 and SINUS-52 were two multicentric, randomized, placebo-controlled trials assessing dupilumab when added to standard care in adults with severe CRSwNP, measuring outcomes at 24 and 52 weeks from the first administration. According to the results that emerged from these trials, dupilumab was proven to reduce NPS, LM score, and severity of symptoms. A safe profile of this mAb was also observed.

Given the strength of the evidence that emerged from these trials, the US Food and Drug Administration (FDA), followed by the European Medicines Agency (EMA) and the Agenzia Italiana del Farmaco (AIFA) recently approved the administration of mAb in severe uncontrolled CRSwNP for the adult population, although FDA sought further clinical data concerning benralizumab, and its final approval is still pending.

Since the approval of dupilumab, mepolizumab, and omalizumab, whenever the disease is scarcely controlled with topical medications or functional surgery, the clinician can rely on a targeted therapeutic approach. The diagnostic workup should be personalized to identify the correct target to treat. Establishing the most appropriate mAb for each single patient will be the next challenge, from the perspective of a personalized approach to the patient. A recent metanalysis by Oykhman comparing each approved randomized controlled trial suggested that dupilumab may be the most effective mAb for CRSwNP, having demonstrated the greatest benefits for all the outcomes studied. However, evidence is still limited and further randomized controlled studies are needed [81]. Other issues are still unsolved: in the next era, research will focus on identifying those biomarkers capable of predicting the response to biologic therapy, as well as on developing other mAbs capable of deactivating the inflammatory process even more selectively.

## 3. Conclusions

CRS is a heterogeneous disease with at least two main clinical phenotypes (CRSwNP and CRSsNP), and it is characterized by a complex interaction between the degree of upper airway inflammation (mainly type-2) and its clinical expression. Genetic, environmental, and behavioral factors, together with the presence of relevant comorbidities, contribute to determine the degree of disease severity (i.e., recurrence rate after surgery and the need for systemic corticosteroid treatment) and its impact on a patient’s QoL and healthcare-related costs (Figure 2). For these reasons, a more personalized approach, including a precise disease endotypization, should be implemented in caring for patients with CRS. In order to achieve this aim, a multidisciplinary team, including at least otolaryngologists, allergologist/clinical immunologists, pneumologists, and phycologists, is advised (Figure 1), particularly in an era in which novel therapeutical approaches, such as biologic agents and innovative surgical treatments (e.g., “reboot surgery”) face the scenario of CRS management.

## Figures and Tables

**Figure 1 jpm-13-00618-f001:**
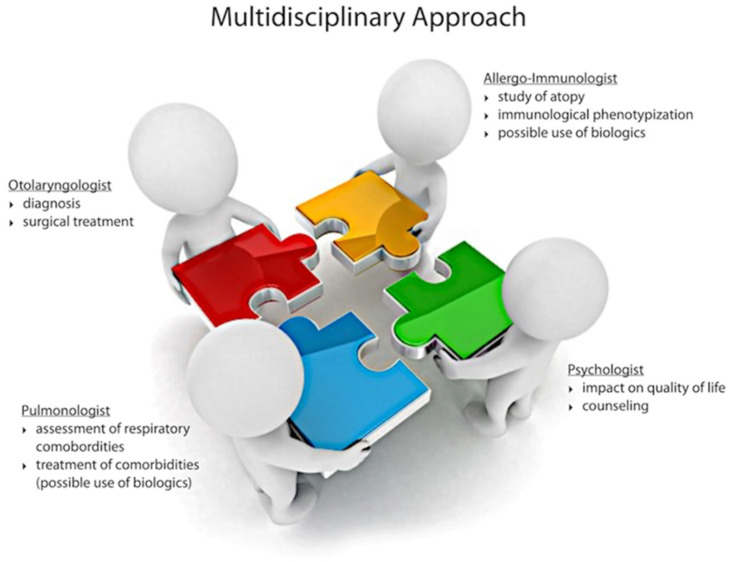
Multidisciplinary approach to patients with chronic rhinosinusitis.

**Figure 2 jpm-13-00618-f002:**
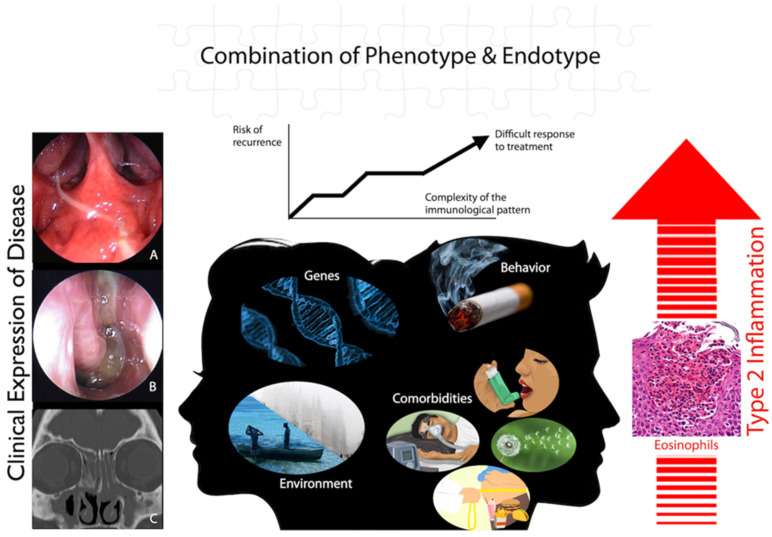
Genetic, environmental, and behavioral factors contribute, together with comorbidities, to determine the degree of upper airway type-2 inflammation, with consequences for clinical expression and severity (e.g., recurrence after surgical treatment) of chronic rhinosinusitis. (**A**) Post-nasal drip, as an example of pure upper airway inflammatory involvement; (**B**) Chronic rhinosinusitis with nasal polyps, a more structured inflammatory condition; (**C**) CT-scan with extended chronic rhinosinusitis with concomitant bone remodeling as a consequence of chronic inflammation.

## Data Availability

Not applicable.

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
