# Peer review of "Care for Patients with Type-2 Chronic Rhinosinusitis"

_jpm, 2023, doi:10.3390/jpm13040618_

Round 1

Reviewer 1 Report

In the article titled “Care about patients with chronic rhinosinusitis type-2 

inflammation-related“ the authors present a multidisciplinary approach for treatment of patients with CRS. In general, the thought of personalized and multidisciplinary approaches in chapter 2 is discussed too little due to no references or own research. Here a more detailed discussion about the benefits and future outcomes is key. 

For other details refer to the comments below.

Chapter 2.2.: 

1)    Line 165-167: Complicated to read, please rephrase. 

2)    Line 191-205: The reference 31 from 1999 represents some revised surgical principles. Nowadays the functional part still should be applicable and achievable for severe CRSwNP. Only in rare cases the functionality criterion cannot be respected. Please rephrase the chapter or give more current references. 

3)    Line 206-209: Why will few patients have access to biologicals? Explain in detail. Also, the fact that surgery will “certainly” be the first therapeutic attempt is under discussion. Please address. 

4)    Lien 323-329: The sentence including the wording “mandatory” not adequate due to the lack of statements/explanations in the previous chapters. 

Reviewer 2 Report

Very good work, however, needs to be supplemented. The authors should note that not only adults but also children suffer from chronic sinusitis. The course of chronic sinusitis in children has its specificity, as seen, among others, in "European position paper on rhinosinusitis and nasal polyps 2020. Rhinology."

Authors should also supplement the literature with Straburzyński M, Gryglas-Dworak A, Nowaczewska M, Brożek-Mądry E, Martelletti P. Etiology of 'Sinus Headache'-Moving the Focus from Rhinology to Neurology. A Systematic Review. Brain Science. 2021 Jan 9;11(1):79. doi: 10.3390/brainsci11010079. PMID: 33435283; PMCID: PMC7827425.; Chmielik LP, Mielnik-Niedzielska G, Kasprzyk A, Stankiewicz T, Niedzielski A. Health-Related Quality of Life Assessed in Children with Chronic Rhinitis and Sinusitis. children. 2021; 8(12):1133. https://doi.org/10.3390/children8121133;Campbell R.G. Pediatric sinonasal surgery: a literature review. Aust J Otolaryngol 2021;4:31.doi: 10.21037/ajo-21-16;Huang Y, Qin F, Li S, Yin J, Hu L, Zheng S, He L, Xia H, Liu J, Hu W. The mechanisms of biofilm antibiotic resistance in chronic rhinosinusitis: A review. Medicine (Baltimore). 2022 Dec 9;101(49):e32168. doi: 10.1097/MD.0000000000032168. PMID: 36626427; PMCID: PMC9750636.

Reviewer 3 Report

Thanks for a great summary of Type 2 CRS.

I would like to make a few comments to improve the quality of the manuscript.

1) Title:
I suggest as ' Care about patients with type 2 CRS.

2) Need references.
    Line 87,  Line 158, Line 224.

3) Line 178 to 183.
  Frontal sinus is large sinus?
  For FESS, maxillary and ethmoid sinus are easy easy to manipulate.
  Is AFRS adequate to control with FESS?

4) 2.3 Are biologics grabbing the spotlight?
  Please review the structure of the paragraph and reorganize it.
  Need to describe the same drug in the same paragraph.

Round 2

Reviewer 2 Report

Thanks to the authors for the changes made. The work is good and interesting. I suggest in the section "Surgical management: contemporary and future perspective" to add information about M.E.Wigand and G . Retinger and the Balloon sinuplasty technique

Good luck